# 8-port MIMO antenna at 27 GHz for n261 band and exploring for body centric communication

**Anupma Gupta[1], Meet Kumari[2], Manish Sharma[3], Mohammed H. Alsharif[4], Peerapong Uthansakul[5]*, Monthippa Uthansakul[5], Shonak Bansal[2]**

**1** Department of Interdisciplinary Courses in Engineering, Chitkara University Institute of Engineering and Technology, Chitkara University, Rajpura, Punjab, India, **2** Department of Electronics and Communication Engineering, University Institute of Engineering, Chandigarh University, Gharuan, Mohali, India, **3** Chitkara University Institute of Engineering & Technology, Chitkara University, Rajpura, Punjab, India, **4** Department of Electrical Engineering, College of Electronics and Information Engineering, Sejong University, Seoul, Republic of Korea, **5** School of Telecommunication Engineering, Suranaree University of Technology, Nakhon Ratchasima, Thailand

* uthansakul@sut.ac.th

**Data Availability Statement:** All relevant data are within the manuscript and its Supporting Information files.

## Abstract

This paper presents a compact 5G wideband antenna designed for body-centric networks (BCN. The single element antenna design includes a simple T-shaped radiator patch with ring shaped ground plane and transformer impedance feedline. First, the antenna was simulated in free-space, and its resonant frequency is found to be 27 GHz, falling within 5G's n261 band. The proposed single radiator antenna has a size of 23.375 mm³, and it offers a wide impedance bandwidth of 2.0 GHz (26–28 GHz). Parametric studies demonstrated that by increasing the length of slots in patch, the antenna frequency can be reduced further. Single radiator antenna is used as 8-element MIMO structure. Parallel adjacent antenna in X-direction has minimal coupling effect, whereas antenna placed in Y-direction has high coupling effect. Thus, coupling is reduced by etching a wall of slots in ground plane. It alters the surface current interference in Y-direction and limits the coupling effect. The antenna is investigated to use in body area network applications. To evaluate its on-body performance, an equivalent body model is virtually developed. The on-body performance is assessed by placing the antenna in close proximity to body model. Stable and robust performance is achieved for the on-body operation. At the resonant point, the antenna exhibits a reflection coefficient of -30 dB (free space) and -40 dB (on-body), high isolation of above 20 dB between adjacent radiators and above 30 dB for other radiators. Antenna has stable performance for different body tissues and on the non-planar structures. Bidirectional radiation pattern with gain of 2.53 dB and broadside type orientations with gain of 4.64 dB are achieved for free space and on body operations respectively. low specific absorption rate makes antenna safe for health care devices. Further, diversity performance is measured in terms of envelope correlation coefficient (ECC), and diversity gain (DG). Maximum Value of ECC is 0.005 and minimum value DG is 9.97 at 27 GHz which confirms the excellence of antenna for MIMO applications.

**Funding:** This work was supported by Suranaree University of Technology (SUT).

**Competing interests:** The authors have declared that no competing interests exist.

## I. Introduction

Body centric communication is an emerging technology with wide range of application including remote health monitoring (e-health-monitoring), implantable devices, smart wearable devices for sports, entertainment and location tracking, and IoMT (Internet of Medical Things). E-Healthcare industry is one of the most rapidly growing industry as society is advancing towards enhancing the quality of life. E-health-care services are implemented through the sensors deployed on/in body organs to transmit various physiological signals wirelessly to the healthcare centers located on different locations. Networking technologies like, Bluetooth, Wi-Fi, internet are required to connect on/in body sensing equipment to the monitoring device and healthcare centers simultaneously [1].

5G technology has the capacity to revolutionize connectivity by enabling massive connections and exceptionally high speeds, thereby facilitating real-time delivery of e-healthcare services. Even a nanosecond delay can have adverse consequences in such scenarios. Notably, 5G proves immensely valuable for the Internet of Medical Things, as it brings forth new medical innovations such as virtual reality, augmented reality, remote patient monitoring, healthcare education, artificial intelligence, and related technologies [2]. Moreover, 5G assumes a pivotal role in the domain of medical IoT, facilitating access to real-time data for instantaneous decision-making. The healthcare system necessitates swift diagnosis and report generation, and 5G ultra-low latency and enhanced computational capabilities expedite communication and enable rapid, accurate diagnoses. Antenna plays a crucial role in wearable networks, and a challenging task for design engineers. Microstrip antenna technology has special benefits in this respect and appropriate for wearable and on body devices due to their peculiar-characteristics of flexibility, comfortability, and lightness [3, 4].

Major properties of body centric antenna that needs to be thoroughly checked includes: wide bandwidth for withstanding frequency detuning affects due to heterogeneous body tissues; conformability for non-planar tissue structures and different body postures and movement; and level of nearfield radiation absorption in terms of specific absorption rate. In spite of aforementioned properties, antenna impedance and surface current characteristics alters on body tissue due to high conductivity and dielectric constant [4]. Thus, it is challenging to attain stable antenna characteristics for body centric communication as compared to free space.

However, even though profound progress has been made for wearable antennas, the pace of developing robust, and stable on-body antenna for advanced communication standards has far lagged behind. EBG structures and AMC backed radiators are widely used to isolate the antenna from body tissue effect and designed using textile substrates to make antenna conformal for bending and non-planar tissue structures [5–7]. In [8], self-grounded technique is adopted for improving radiation efficiency and bandwidth. Surface integrated technology is looked into [9] for shielding the back scattered waves by body tissue. However, these topologies are inherently complex, suffers with higher losses due to soldering and occupies larger space. Besides this, use of fabrics and other elastrorous substrates reduces the robustness and durability of components and sensitive to humidity and temperature. In order to design the planar and simple structures with wider bandwidth CPW fed antenna with a metallic back reflector is proposed in [10]. Defected ground structure is used in [11] to enhance the impedance bandwidth and multiple slots and open ended slits are etched in the radiator for miniaturization [12]. Furthermore, a study put forth a Hexagonal Fractal Antenna Array (HFAA) aimed at advancing next-generation wireless communication [13]. In the realm of wearable technology, a disk-shaped antenna operating on electromagnetic coupling principles between feed and radiator elements was proposed [14]. Additionally, a textile antenna operating at 26/28 GHz, exhibiting a gain of 7 dB, was proposed in [15]. An innovative dual-meander-line

antenna was introduced, capable of spanning multiple bands (58/44/34/22 GHz) to broaden bandwidth and increase the number of resonance bands, thus enhancing performance [16]. Although, wider bandwidth and multiband response has been attained in plethora of papers, a significant shifting of resonance frequency has been observed due to structural deformation and body postures. In addition to this, restrictions of limited radiation power in body centric communication, and multipath fading effects caused by internal scattering/reflections of heterogeneous body tissue layers dramatically diminish the reliability/performance and robustness of communication [17–20]. Therefore, diversity technique MIMO is attaining much interest in body area networks. A very few literature is available for wearable MIMO antenna structures for 5G communication [17, 19–24]. Different techniques for enhancing the diversity performance is presented by the researchers. In [17], a mm wave 2-port MIMO antenna with sectoral slots on circular patch is investigated. Planar CPW fed antenna for UWB MIMO operation is proposed in [18] for wearable applications. A large decoupling network with defected ground structure is incorporated to enhance the port isolation. CPW fed 4-port antenna structure is investigated in [19], decoupling network is integrated with unconnected ground substrate. A compact and flexible wearable antenna is designed and evaluated for 5 G MIMO applications [20]. In [22], self-isolated, complementary split resonator technology is used. Most of the literature presents complex geometries for enhancing port isolations, which unnecessarily increases the overall size of the MIMO structure.

New wearable MIMO antennas with solution of all aforementioned issues are of great interest for researchers [25]. Proposed structure has the features of compact and simple geometry, stable impedance and radiation characteristics for heterogeneous body tissues, conformal to use with wearable devices, ensure tissue safety from EM waves, excellent diversity performance for MIMO operation. In this paper, a compact, Slotted ground MIMO antenna structure has been proposed for both the free space and on-body devices operating at 27 GHz for 5G new radio bands lying in frequency range2 (FR2). Initially, a compact antenna is designed with defected ground structure at required operating frequency in free space. Then 2-port antenna is designed for MIMO application and verified for on-body application. To design the larger MIMO structures, 2-port structure is converted into 4-port MIMO by using the complementary symmetry configuration of 2-port structure along Y-axis. In this configuration antenna lost its impedance matching and have high mutual coupling among radiators. Thus, ground plane of antenna is modified by etching port of slots between radiators place in Y-direction and mutual coupling is reduced. Finally, 8-port port structure is designed, numerically evaluated for both the free space and on-body environment, fabricated and tested. Computer Simulation Technology (CST) Microwave Studio Suite, a reliable electromagnetic simulation program, is used for the antenna's design and simulation.

## II. Single radiator antenna design methodology in free space

Design topology for the single radiator of proposed antenna is shown in Fig 1. Initially, single radiator structure with defected ground technology is designed to attain the resonance bandwidth for n261 band. Transformer impedance feedline is considered for port impedance matching with thin substrate of 0.5 mm thickness. Substrate material Fr4 of relative permittivity 4.2. Planar dimensions of single radiator structure are 5.5 mm ×8.5 mm. Antenna is designed using a rectangular-ring shaped defected ground and a rectangular radiator with two parallel slots. Detailed design process is evaluated in three steps as shown in Fig 2. Reflection coefficient plot for the step wise structures is shown in Fig 3.

In first step, conventional patch antenna with full ground plane is considered that is resonating at 34 GHz with reflection coefficient of -14 dB (Fig 3). Surface current distribution is

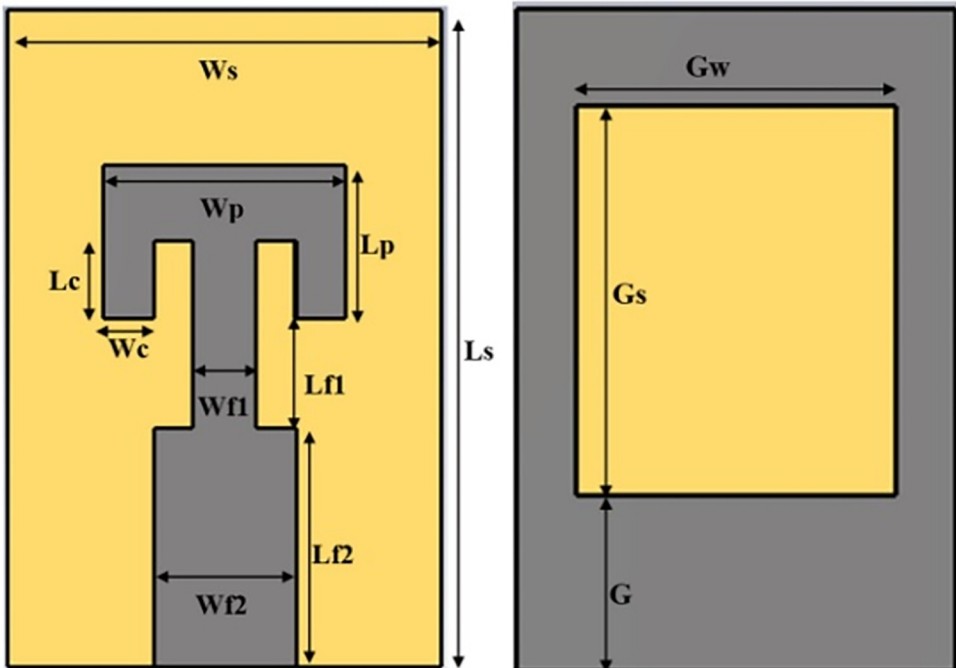

**Fig 1. Design topology of single radiator.** (A) front view (B) back view.

analyzed to observe antenna behavior and represented in Fig 4. In order to lower the resonance frequency towards 27 GHz, ground plane is modified as ring shaped ground. It alters the resonance path of antenna current and changes the frequency. Ring shaped ground increases the inductance of antenna. To justify it, impedance plot is represented in Fig 5. Impedance plot shows that reactance plot for both the step1 and step2 is approximately overlapped. Resistance value in real impedance plot is also have equal magnitude for step1 and step2. Resonance frequency is lowered and shifted to 30 GHz as shown in Fig 3. It reveals that xL (inductive impedance) is constant for both the steps, L (inductance) is increased which in turns lowers the resonance frequency (f) in step2 (as xl = $2\pi fL$). Further in step3 to attain desired resonance at 27 GHz, surface current length is increased by etching 2-parallel slots in the patch. Surface current plot for step3 make it clears that after etching slots resonance

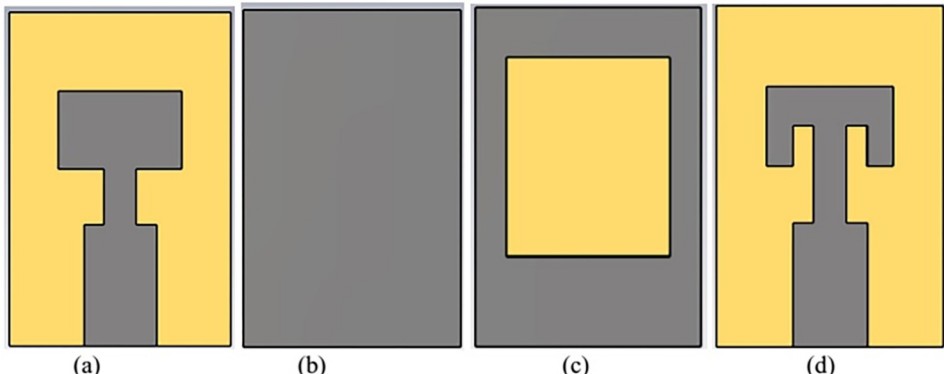

**Fig 2. Step wise geometry of the designed structure.** (A) front view for step1 and step2 (B) back view step1 (C)) back view step2 and step3 (D)) front view step3.

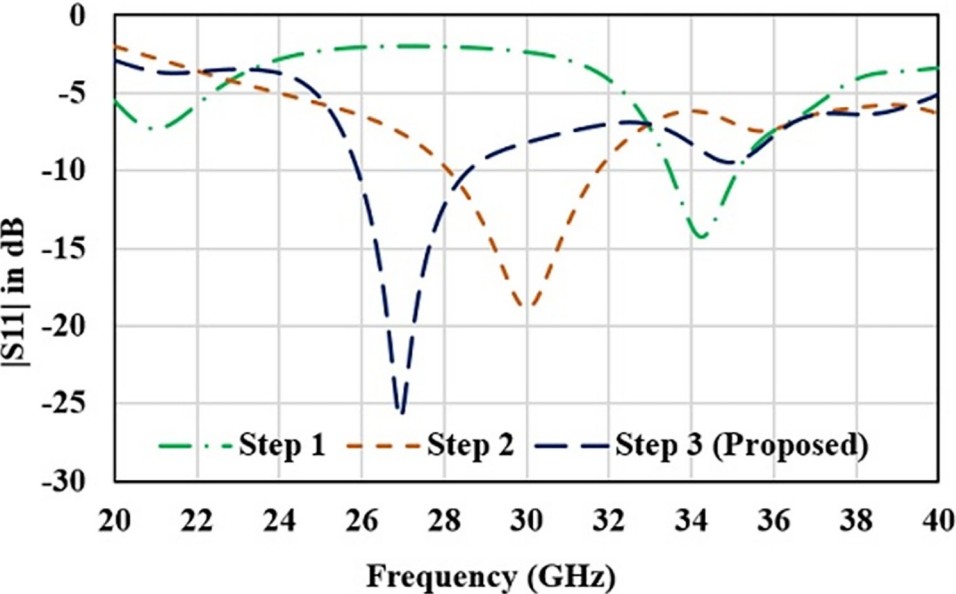

**Fig 3. Reflection coefficient for step wise antenna topologies.**

current length at radiator is increased. From the impedance plot it can be found that 50-ohms resistance with null reactance is attained at 27 GHz. Due to etching of slots, antenna parallel capacitance (C) is increased which in turns reduce the capacitive reactance (Xc) of antenna and resonance frequency (as C = 1/2πfXc). Shifting of the real and imaginary impedance curve in step3 justify the impact of slot on antenna performance. Impact of varying slot length (Lc) on antenna resonance frequency is represented in Fig 6. Value of Lc is varied from 0.5 mm to 1.2 mm. With the increasing slot length resonance frequency is decreasing. Desired frequency is obtained at slot length of 1.0 mm.

## III. MIMO antenna configurations

Single radiator antenna is configured as MIMO antenna to enhance the communication channel capacity. First a dual pair is designed by placing the identical antenna pair parallel to each other as shown in Fig 7. S-parameters plot for the 2 element MIMO configuration is shown in Fig 8. It is clear that stable resonance frequency is achieved with the connected ground MIMO structure. Here S11 = S22 and S12 = S21 depicting negligible impact of two radiators on each other. S21 value is also below -20 dB depicting high port isolation between two radiators.

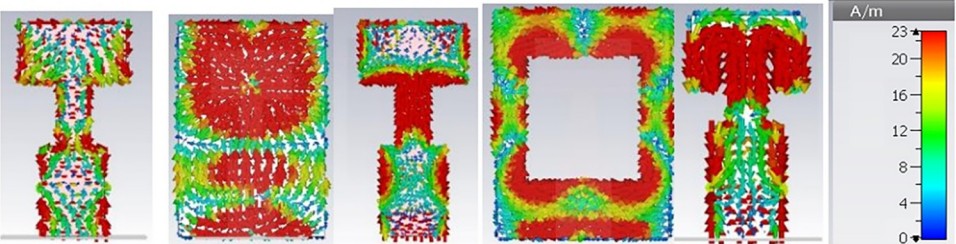

**Fig 4. Surface current distribution.** (A) step1 front view (B) step1 back view (C) step2 front view (D) step2 and step3 back view (E) step 3 front view.

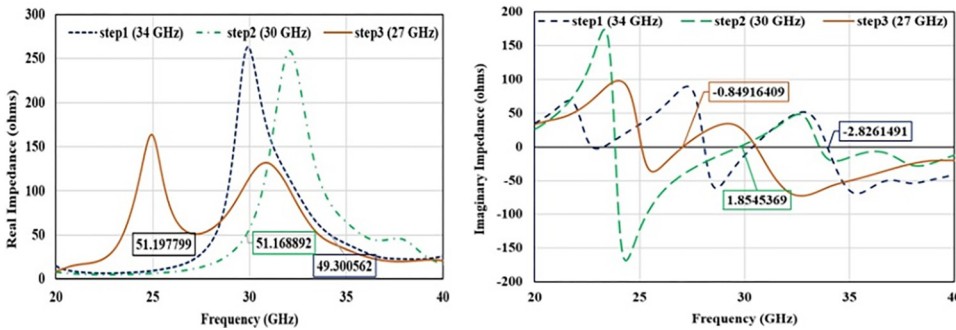

**Fig 5. Impedance plot.** (A) real impedance (B) imaginary impedance.

Surface current distribution in Fig 7C depicts that very low power is coupled form antenna1 to antenna2.

To further enhance the channel capacity 2-element MIMO configuration is converted into 4-element structure. Dual radiator configuration is placed in complementary symmetry configuration along Y-axis as shown in Fig 9A and 9B. S-parameter plot for this configuration is plotted in Fig 10. It shows that antenna frequency band detuned to 30.5 GHz for ant1 and ant2 while no resonance band it attained for ant3 and ant4. Mutual coupling for ant1 and ant2 is also very high. It is probably due to the connected ground structure; common-ground for such small structures work as a radiating element and allows surface waves to pass in between the radiators. It interferes with the surface waves of the neighboring elements, and enhances the coupling effect. Thus, a decoupling technology is required to make antenna suitable for large array MIMO structure. Surface current distribution of 4-element antenna without the decoupling structure in Fig 11A. It depicts that current is oriented towards ant1 to ant2 through connected ground plane in vertical direction. In order to delimit the current flow, array of parallel slots is etched in the ground plane along the connected surface as shown in Fig 9C. This defected ground structure act as a wall between the radiators placed in complementary symmetry contributions and alters the surface wave path. Surface current distribution plot for the

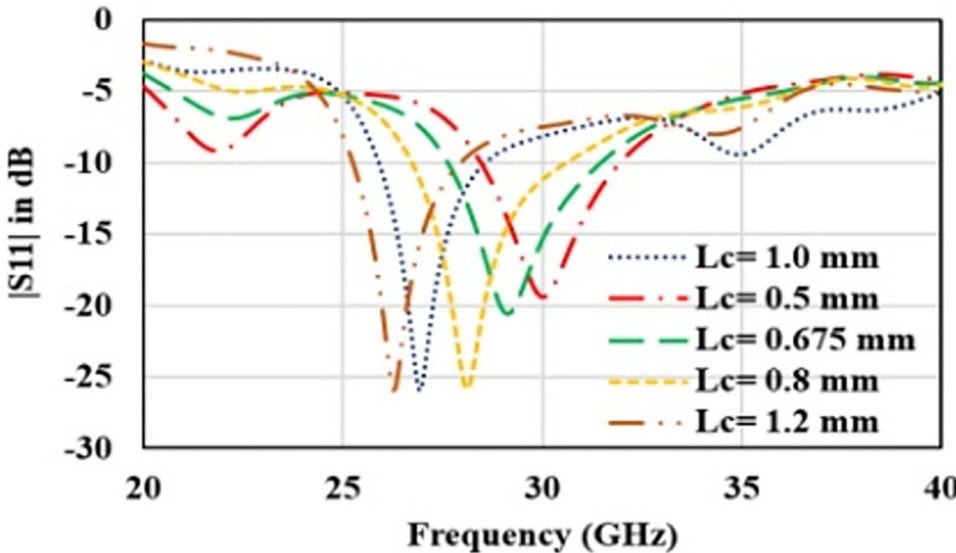

**Fig 6. Reflection coefficient plot for patch slot length 'Lc'.**

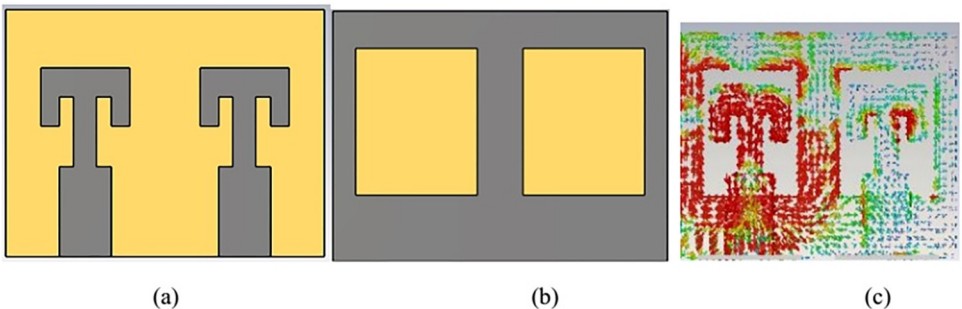

**Fig 7. 2-element MIMO antenna.** (A) Front View (B) Back view (C) Surface current.

defected ground MIMO structure is represented in Fig 11B. It can be found that negligible current is oriented towards the neighboring elements. Geometrical parameter details of the antenna marked in Figs 1 and 9C are listed in Table 1.

S-parameter plot for 4-element MIMO antenna with defected ground decoupling structure is shown in Fig 12 (reflection parameters) and Fig 13(transmission parameters). It ensures that after etching the slot walls, resonance frequency, impedance bandwidth is stable and mutual coupling is below -20 dB for all the radiators.

Furthermore, antenna performance is analysed by deploying the designed configuration in 8-port MIMO antenna as 2×4 array as shown in Fig 14. Reflection and transmission coefficient plot for 8-element structure are shown in Figs 15 and 16 respectively. Reflection coefficient plot for all the radiators has effectively covered the desired bandwidth from 26 GHz to 28

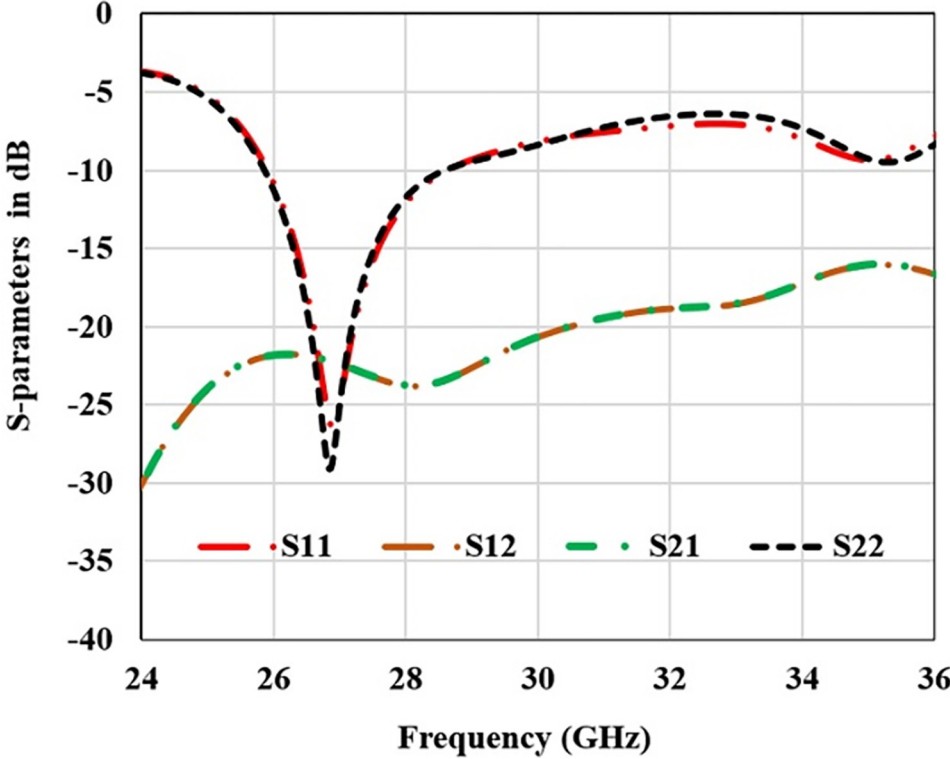

**Fig 8. S-parameters plot for 2-element MIMO structure.**

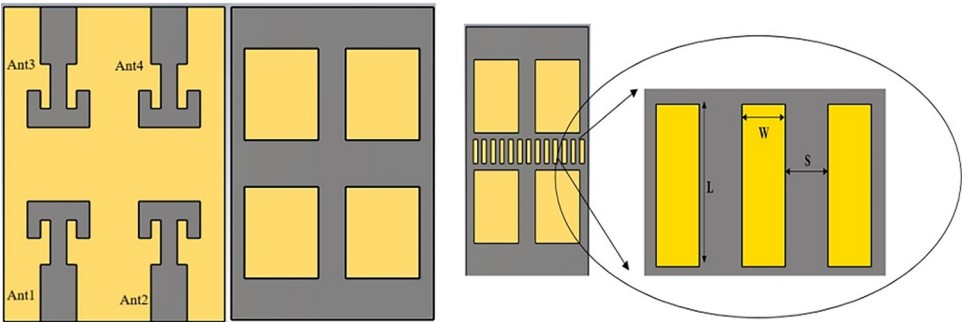

**Fig 9. 4-element MIMO structure.** (A) Front view(B) Back view without decoupling structure (C) Back view with coupling structure.

GHz. Transmission coefficient curves for port1 ($S_{n1}$) are shown as curves for other ports are approximately overlapping. Maximum coupling of -21 dB is occurred due to the parallel adjacent radiator ($S_{21}$) and minimum coupling of -50 dB is achieved between the complementary symmetrical placed antenna ($S_{31}$).

## IV. Antenna performance for on body operation

To make antenna suitable for commercial and real time applications, performance of 2-Port MIMO structure is studied for on-body devices. Human tissue is a complex structure with heterogeneous dielectric properties. Behavior of antenna may alter due to close proximity of body tissue. Thus, different tissue phantoms are designed to test the antenna robustness. Fig 17 shows the three layered rectangular, 3-layered circular, and a voxel arm phantom models used for setting up the on-body simulation setup. Three layers of simplified phantom model consist of skin (3 mm), fat (5 mm) and muscle tissue (12 mm). Planar dimensions of all layers is 40×40 mm$^2$. Electric properties of tissue layers are considered according to [26, 27]. Human body is a non-planar structure and undergoes various movements and postures. Thus, wearable devices need a flexible antenna with stable impedance matching and resonance frequency.

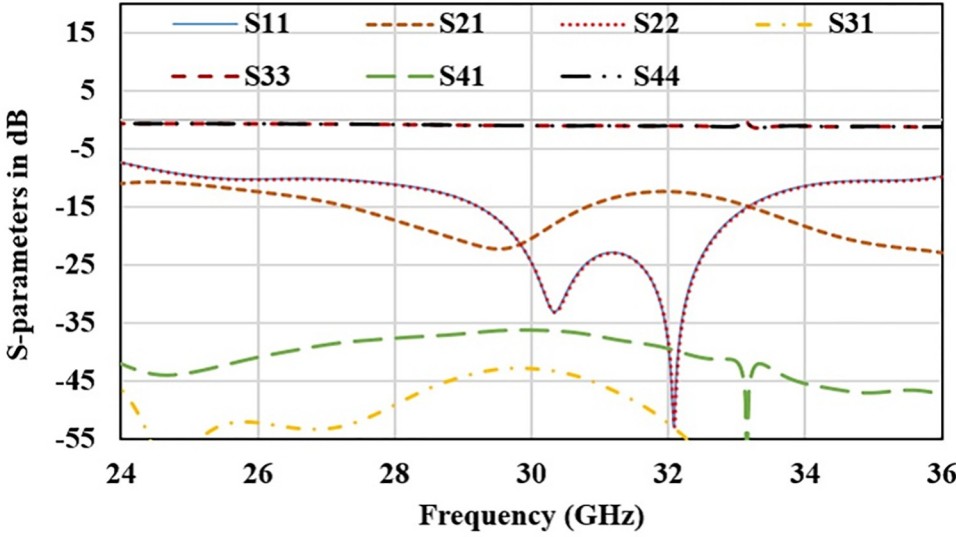

**Fig 10. S-parameter plot of 4-port MIMO antenna without decoupling structure.**

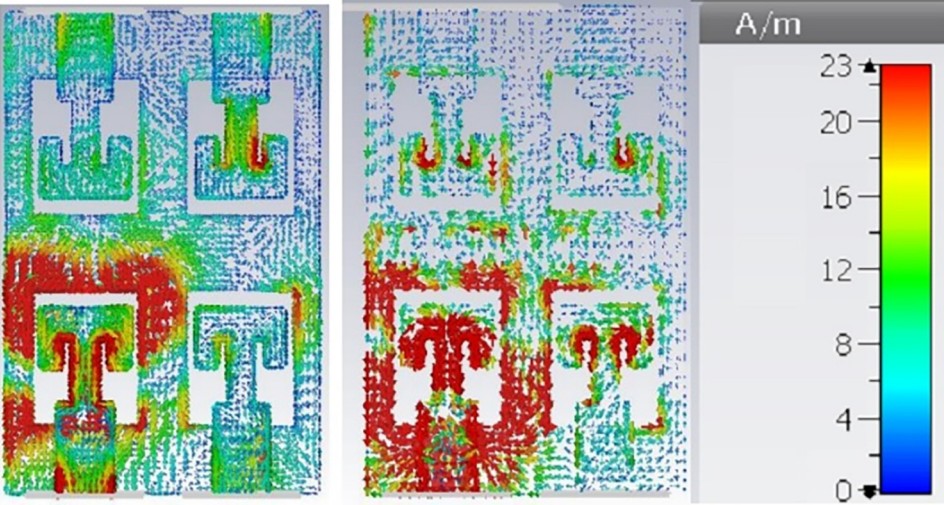

**Fig 11. Surface current distribution of 4-element structure.** (A) without decoupling structure (B) with decoupling structure.

Cylindrical phantom is considered to test the antenna performance for the bending condition. Cylindrical phantom has radius of 18 mm, outer layer of skin tissue with radius 18 mm, middle layer is fat with radius of 15 mm and innermost layer is of muscle tissue with 10 mm radius. Antenna is bent across the cylindrical phantom at the radius of 18 mm, which is taken as the average of the human arm. Fig 18 represents the S-parameters curve for all the three on-body operating conditions. For the flat rectangular surface antenna has the good impedance matching and covered the desired band from 26 GHz to 28 GHz. Whereas in the bent state and on complex voxel arm reflection curve shifts slightly upward and impedance bandwidth is broadened. On the complex tissue layers, internal reflection of electromagnetic waves takes place, that interfere with the antenna near field radiations. It reduces the quality factor of antenna and in turns increases the antenna bandwidth. Even though antenna has excellent performance on body-tissue with reflection coefficient of -20 dB. Transmission coefficient for all the three operating conditions is also below -20 dB, ensuring good MIMO performance for on-body applications.

## V. Result analysis

The proposed 8-ports antenna is fabricated and simulated results are verified with the measured data for both the free space and on body operations. Agilent N5247A programmable

**Table 1. List of geometrical parameters.**

| Parameters | Value (mm) | Parameters | Value (mm) |
|---|---|---|---|
| Ws | 5.5 | Ls | 8.5 |
| Wp | 3.06 | Lp | 2.0 |
| Wc | 0.63 | Lc | 1.0 |
| Wf1 | 1.8 | Lf1 | 3.1 |
| Wf2 | 0.8 | Lf2 | 2.4 |
| Gw | 4.0 | Gs | 5.0 |
| G | 2.25 | L | 0.8 |
| W | 0.35 | S | 0.46 |

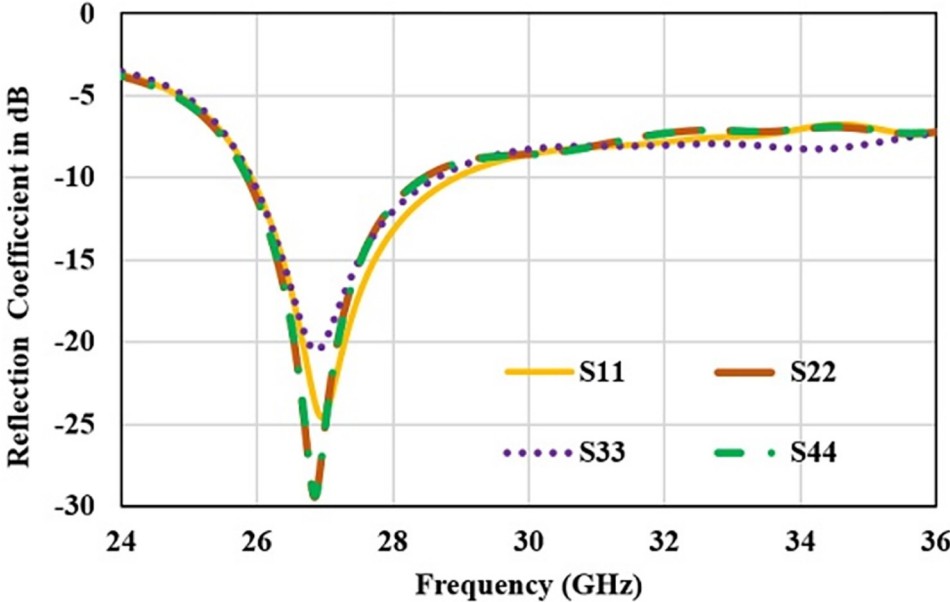

**Fig 12. Reflection coefficient for decoupled 4-element MIMO antenna.**

network analyzer and anechoic chamber is used to measure antenna parameters. To measure on-body characteristics of antenna, antenna is placed on a pork lion. Photographs of the simulation and experimental setup for 8-ports antenna are shown in Fig 19.

S-parameters are the important to analyze impedance matching, bandwidth and mutual coupling among the radiators. Fig 20 represents the simulated and measured S-parameters in free space scenario for port2, port4, port5 and port7. Port1, port3, port6, and port8 are

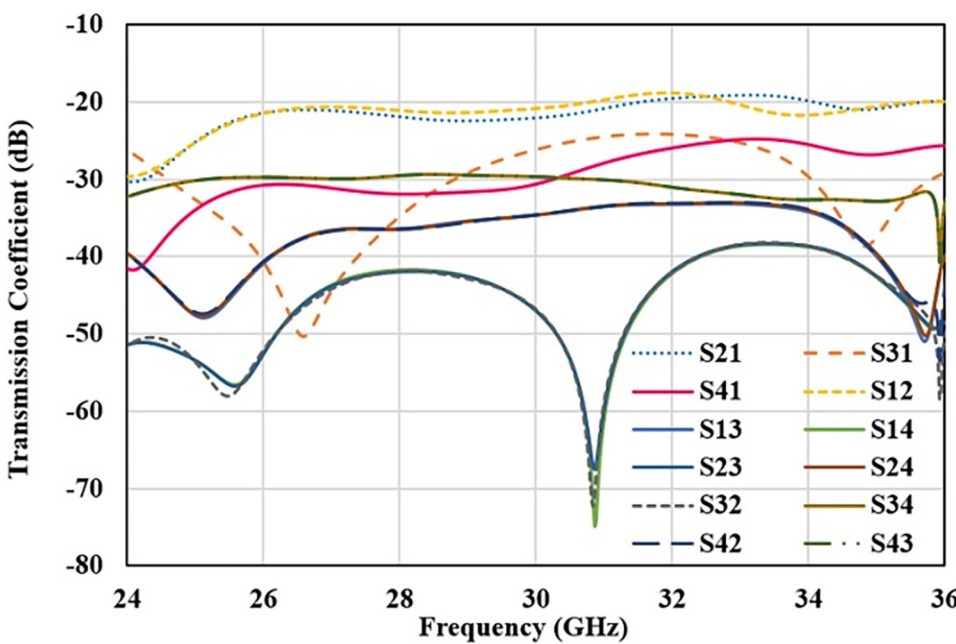

**Fig 13. Transmission coefficient for decoupled 4-element MIMO antenna.**

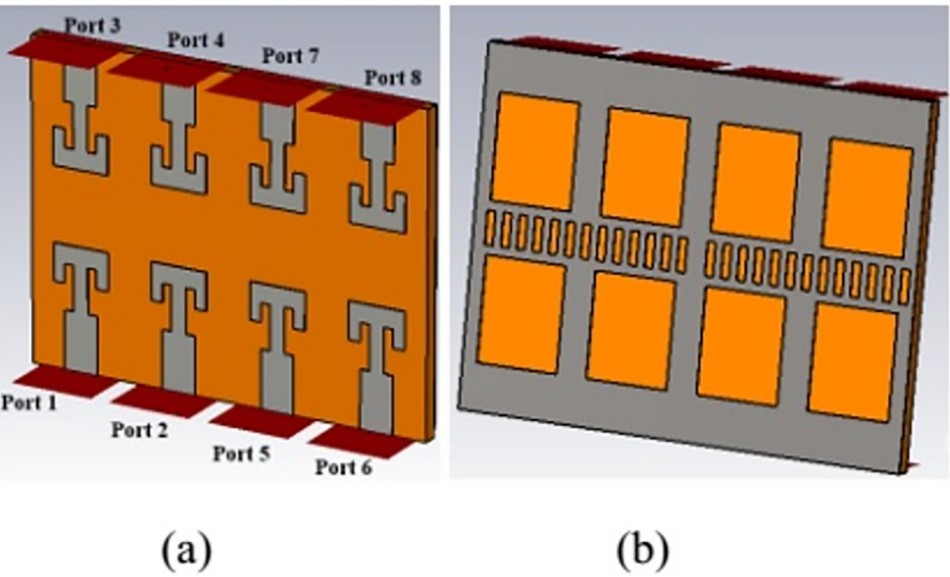

**Fig 14. 8-port antenna array.** (A) Front view (B) Back view.

approximately overlapped to shown curves and for better readability curves for only 4-ports are shown. Reflection coefficient depicts that the operating bandwidth for all the radiators is about 2 GHz (from 26 GHz to 28 GHz) covering desired n261 band for 5G communication. Transmission coefficient for the operating bandwidth is below -20 dB for all the ports. Maximum coupling of -21 dB is obtained in the parallel adjacent port 21, and reduces to -35 dB to the diagonal opposite ports 8 and 1. It Shows that antenna is well suited for MIMO application. Fig 21 shows the S-parameters for on-body scenario for the similar ports as shown for free space. Reflection parameter is below -40 dB as compared to -30 dB in free space, -10 dB

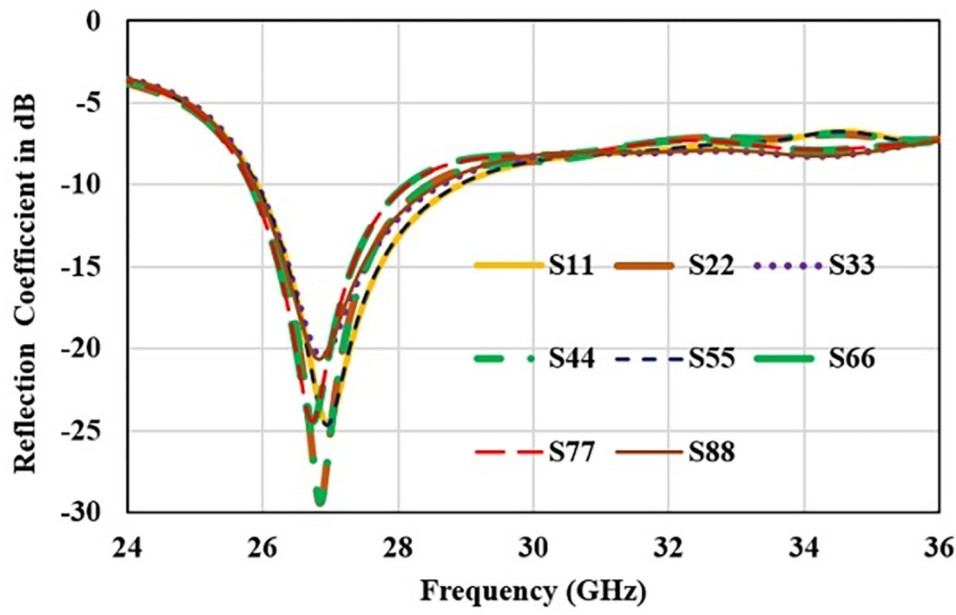

**Fig 15. Reflection coefficient for 8-elemnet antenna.**

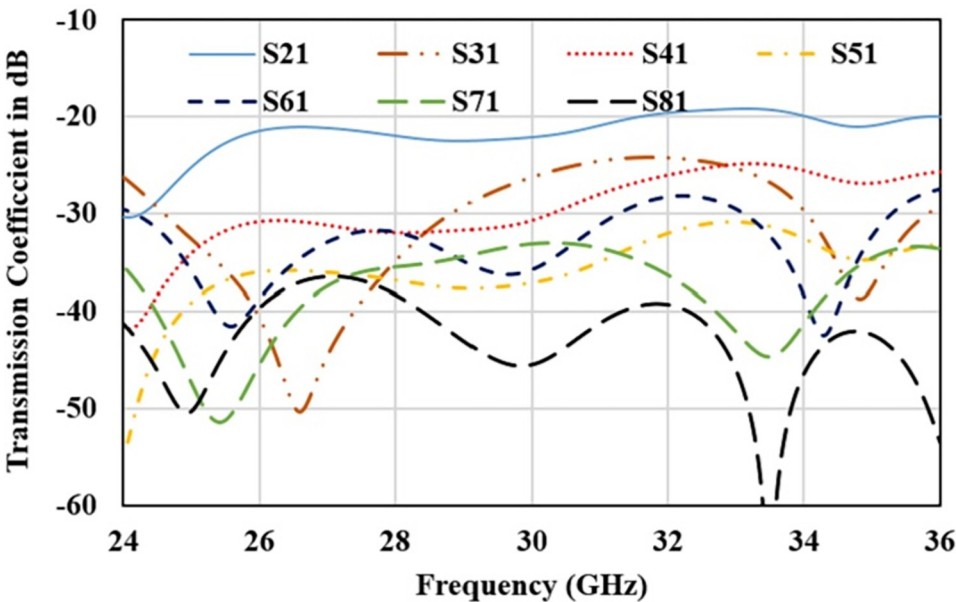

**Fig 16. Transmission coefficient for 8-element antenna.**

impedance bandwidth is also 2 GHz and effectively occupied the desired band. Variation in reflection is due to the internal reflections caused by the interface of low and high dielectric layers of antenna and body tissue. Similar to the on-body performance, port coupling is below -20 dB for all the radiators and minimum coupling of -40 dB is achieved between port 8 and port 1.

Fig 22 shows the 2-D radiation pattern of antenna for free space and on body operating conditions at 27 GHz. Dipole-like radiation pattern has been observed for free space and broadside radiation pattern is observed for on-body condition in both the E-plane and H-plane. On-body operation, internally reflected and refracted waves superimposed with the antenna radiations and increases the forward radiation of the antenna. Broadside radiation pattern is desirable for the on body devices for making reliable communication link with the external monitoring devices and reducing the absorption of EM waves by body tissue. It is virtually impossible to maintain same orientation with on and off body device due to different human activities. Therefore, broadside radiation pattern helps to overcome orientation mismatch and transmits maximum energy in the direction opposite to body. Simulated and measured results are in close approximation to each other. Both in E-plane and H-plane maximum

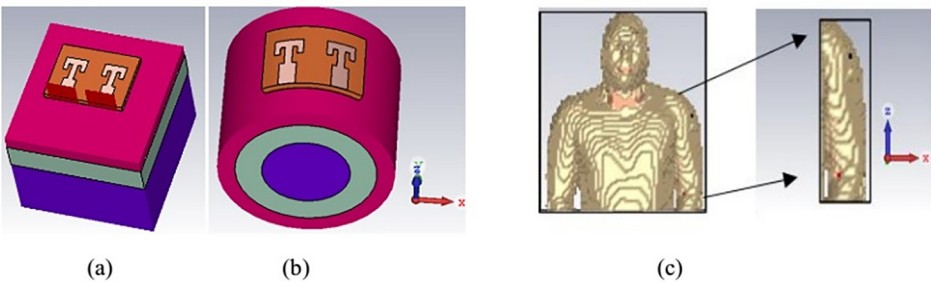

**Fig 17. On-body simulation setup.** (A) rectangular phantom (B) circular phantom and (C) voxel arm model.

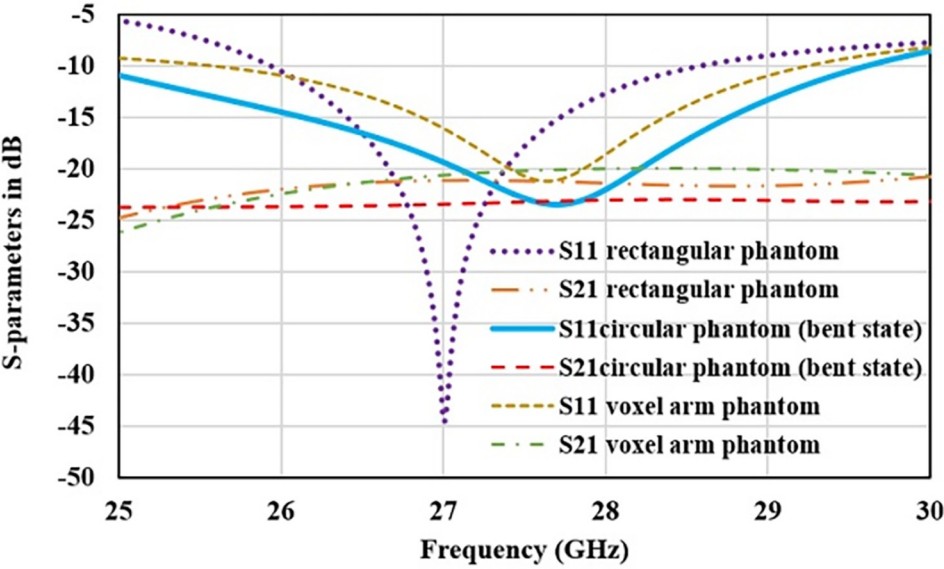

**Fig 18. S-parameter curves of antenna with body tissue phantoms.**

radiation intensity can be observed at 30° and 315°. 3-D power pattern for the antenna is shown in Fig 23. Maximum free space gain is 2.53 dB and on-body gain is 4.64 dB. As back radiations are reduced in the on-body operations, antenna gain is improved in forward direction. Reflection coefficient of proposed structure is also improved for on-body configuration and justify the antenna on-body performance. Besides this, directivity of antenna is one of the important characteristic for the on-body devices. Fig 24 shows the 3-D directivity plot for the free space and on-body operation and confirming a forward-oriented beam suitable for on body to off body communications. Maximum directivity of 5.06 dBi and 9.5 dBi is achieved for free space and on-body conditions respectively. High directivity shows that antenna radiation intensity is higher in a specific direction. Maximum radiations are oriented towards the normal to the surface of body tissue, thus reducing the absorption and loss of radiations in body tissue. Improvement of directivity for on-body condition is due to internal reflections by the heterogeneous body tissue layers.

To support the MIMO capability of proposed antenna, important diversity parameters envelope correlation coefficient (ECC), and Diversity gain (DG) are analyzed as explained in

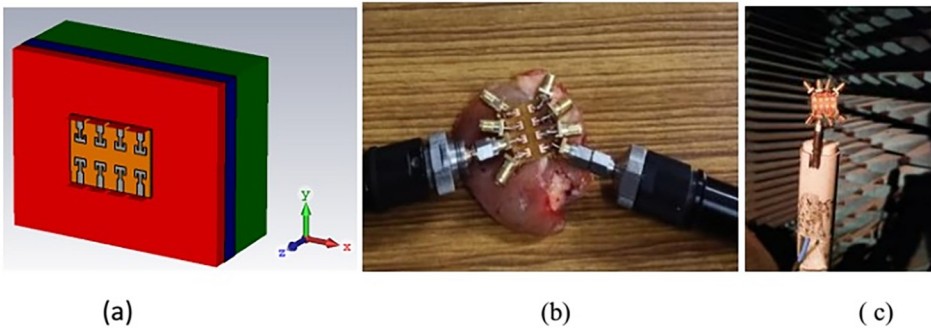

**Fig 19. 8-ports antenna.** (A) on-body simulation setup for 8-port antenna (B) on-body measurement setup (C) antenna in chamber for radiation pattern measurement.

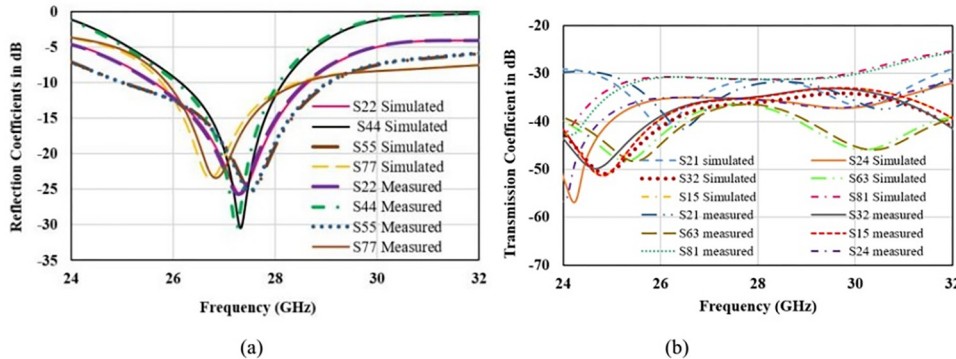

**Fig 20. Free space S-parameters.** (A) Reflection coefficient (B) Transmission Coefficient.

[23, 28]. ECC signifies the correlation of antenna radiation pattern with other antenna when operated in closed proximity. High value of ECC degrades antenna MIMO performance. ECC value lower than 0.05 is acceptable for MIMO antennas. ECC is calculated using 3-D radiation pattern as

$$\rho_e = \frac{|\oiint_\Omega [\overrightarrow{E}_1(\theta, \phi) \cdot \overrightarrow{E}_2^*(\theta, \phi)]d\Omega|^2}{\oiint_\Omega |\overrightarrow{E}_1(\theta, \phi)|^2 d\Omega . \oiint_\Omega |\overrightarrow{E}_2(\theta, \phi)|^2 d\Omega} \tag{1}$$

Where, $\overrightarrow{E}_1(\theta, \phi)$ and $\overrightarrow{E}_2^*(\theta, \phi)$ are the far fields of antenna for the radiator1 and radiaor2.

Diversity gain shows accomplishment of diversity performance of MIMO antenna and it can be evaluated using ECC.

$$DG = 10\sqrt{(1 - |\rho_e|)} \tag{2}$$

Simulated and measured ECC and DG plots for free space and on-body scenarios are shown in Fig 25. It can be observed that ECC results are well below 0.006 and DG is more than 9.95 dB for the operating bandwidth in free space and below 0.005 and DG is more than 9.98 dB for on-body operation.

Specific absorption rate is also the most important parameter for on-body applications. Human tissue absorb the electromagnetic radiations and causes heating effect, it may cause the damage of tissues and affects blood circulation. SAR value gives the limit of heat exposure by

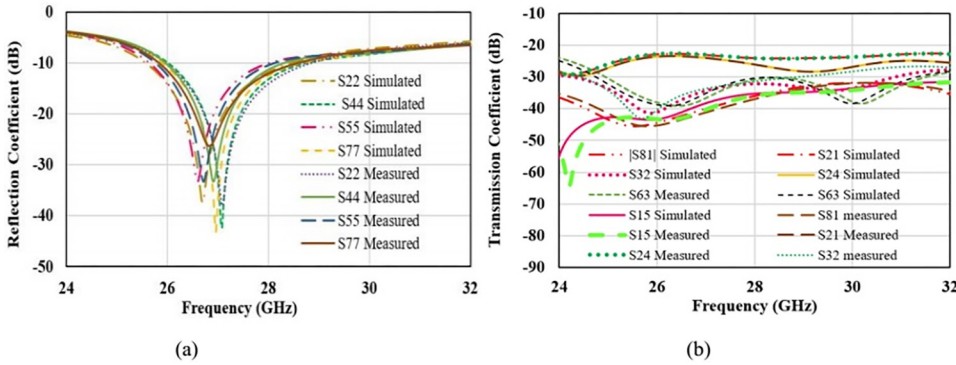

**Fig 21. On-body S-parameters.** (A) Reflection coefficient (B) Transmission Coefficient.

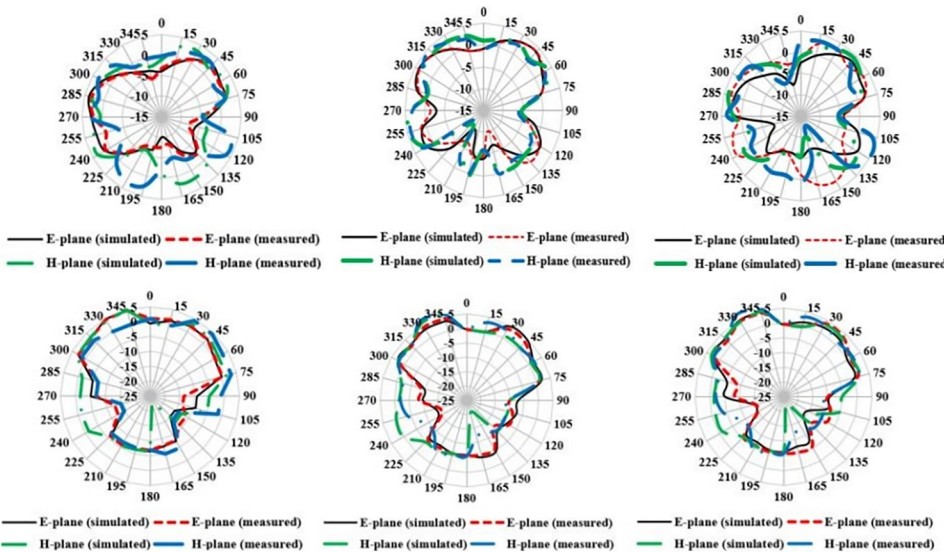

**Fig 22. Radiation characteristics at 27 GHz.** (A) free space port1 (B) free space port3 (C) free space port6 (D)on-tissue port1 (E) on-tissue port 3 (F) on-tissue port 6.

radiating devices. To ensure the safety of user, SAR value is numerically simulated and shown in Fig 26. IEEE C95.1–1999 standard limit the SAR value at 1.6 W/Kg. for 1-gram of body tissue. Proposed structure has SAR value of 0.00427 W/Kg, 0.00123 W/Kg, and 0.00357 W/kg for

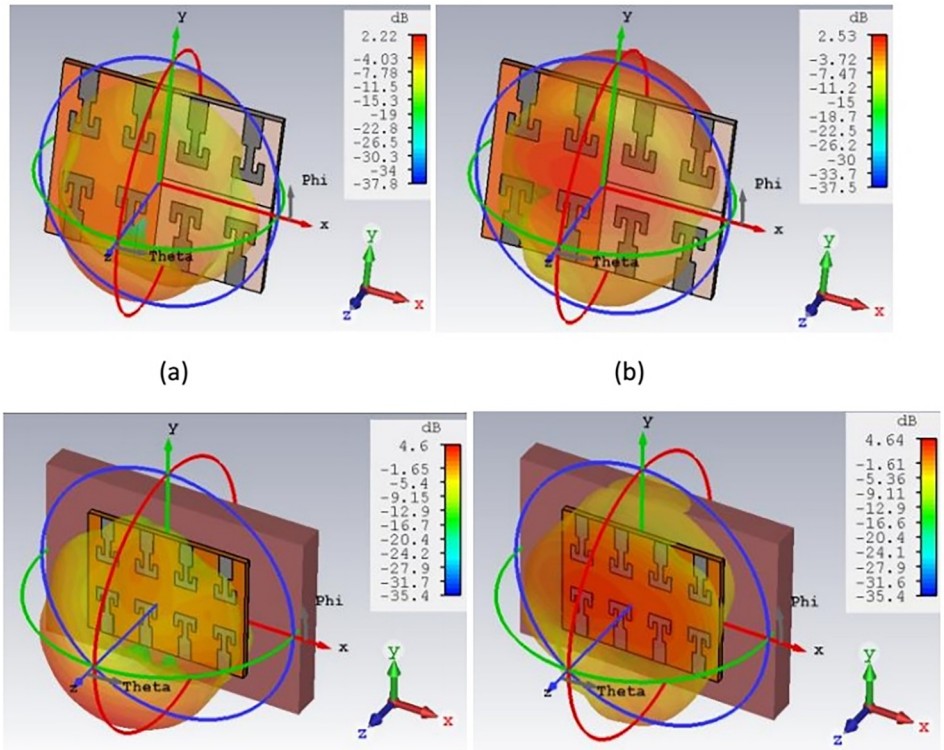

**Fig 23. 3-D power pattern.** (A) free space for port-2 (B) free space for port-8 (C) on-body port-2 (D) on-body port 8.

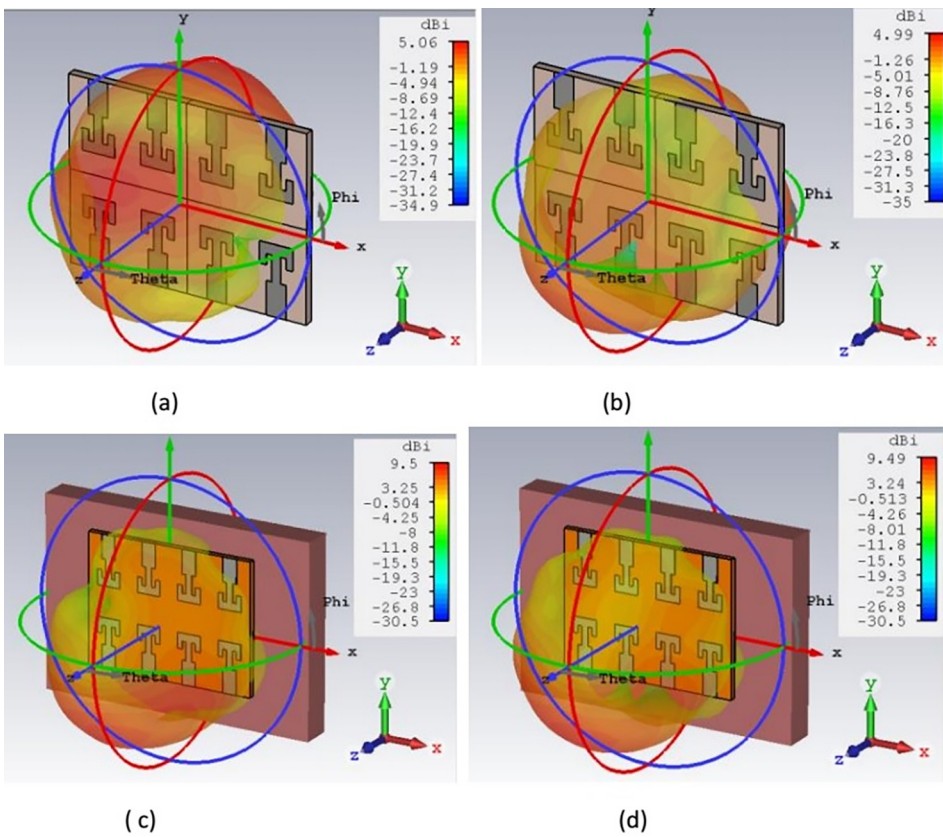

**Fig 24. 3-D directivity plot.** (A) free space for port-2 (B) free space for port-8 (C) on-body port-2 (D) on-body port 8.

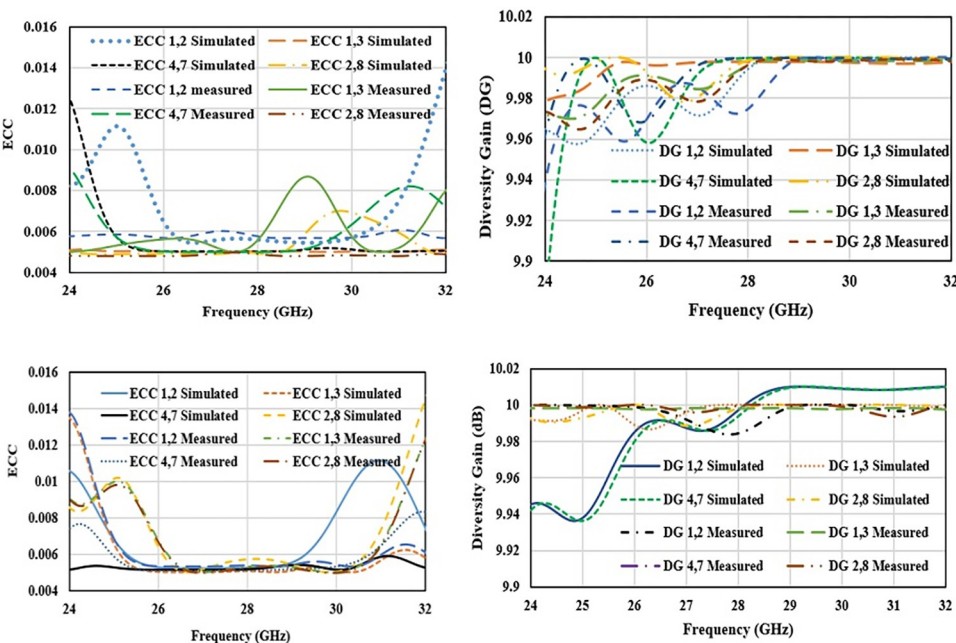

**Fig 25. Diversity performance.** (A) free space ECC (B) free space diversity gain (C) on-body ECC (D) on-body diversity gain.

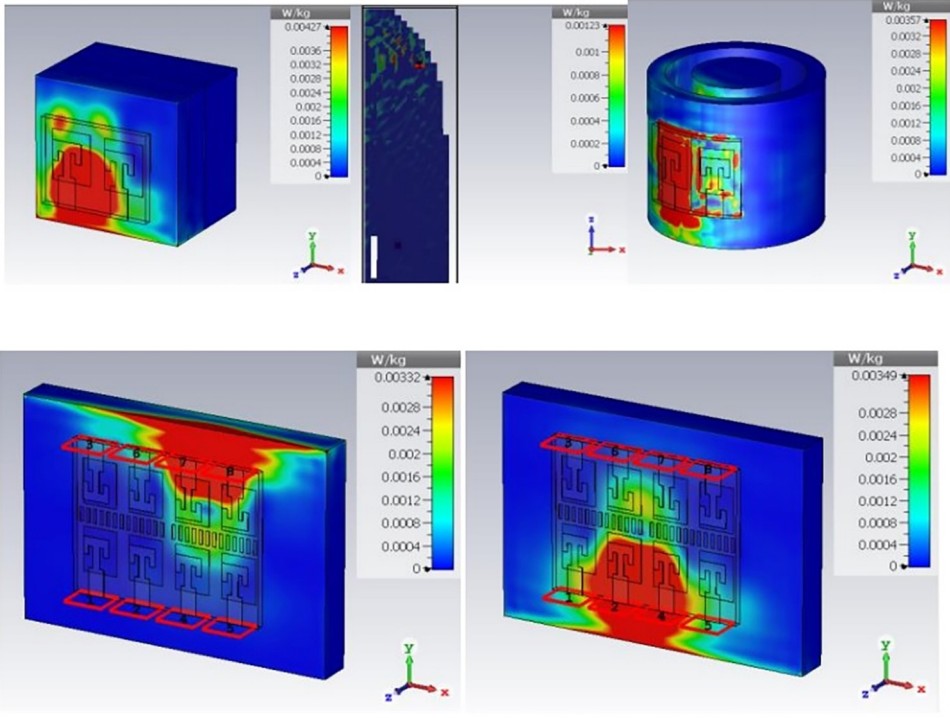

**Fig 26. Specific absorption rate.** (A) on rectangular phantom (B) on voxel arm (C) on circular phantom in bent state (D) SAR for 8-port structures for radiator 2 and radiator 8.

rectangular, voxel arm, and circular phantom models. For 8-port MIMO antenna SAR is evaluated for port2 and port 8 which is 0.00332 and 0.00349 W/Kg. respectively. Input power for SAR calculation is considered as 0.01mW. Obtained SAR value is far below the safety limits and make antenna safe for body centric communication. Table 2 presents a comparison between the current contribution and previous works in the field. Present structure has compact and simplest geometry for 8-element structures, high port isolation, excellent radiation characteristics, secure for on body operating condition with stable performance.

## VI. Conclusions

A novel 8-port MIMO patch antenna is designed and evaluated for free space and body centric applications. The antenna structure, operating principle, parametric study, and experimental results for the proposed antenna have been addressed. Antenna is centered at 27 GHz and occupies frequency spectrum from 26 GHz to 28 GHz and suitable for 5G NR FR2 n261 band. Antenna has a compact and simple geometry and presents a novel approach to reduce the mutual coupling among the radiators. For on-body performance, antenna is numerically simulated on different types of tissue layers including rectangular, voxel arm, and circular phantom for bent state. Antenna has good frequency and impedance characteristics with Low SAR on all the simulation setups. A prototype of the antenna has been fabricated and tested for free space and on-pork lion tissue. The proposed antenna has achieved a gain of 4.64 dB, SAR value of 0.00427 W/Kg. for .01 mW of input power, and port isolation of more than 20 dB. ECC and DG have values of 0.005 and 9.97 respectively at 27 GHz. The present structure has significant performance for 5G MIMO applications both in free space and on-body wireless systems.

**Table 2. Comparison of antenna performance with existing literature.**

| Ref. | Size (mm × mm) | Frequency (GHz) | Directivity | SAR (W/Kg.) | Structure | Port Isolation (dB) | MIMO Array | Material used | ECC | DG (dB) |
|---|---|---|---|---|---|---|---|---|---|---|
| [2] | 14×10.5 | 60 | 10.6 dBi | NA | Parasitic patch loaded Square ring radiator | NA | Single radiator | Flexible printed Board | NA | NA |
| [10] | 29.08×11.42 | 28 | 11.7 dBic | 1.15 | CPW fed with below ground | NA | Single Radiator | RO 4003C | NA | NA |
| [14] | 31×15.10 | 61 | 5.2 dBi | NA | Surface integrated waveguide feed with circular radiator | NA | Single Radiator | RT Duroid 5870 | NA | NA |
| [17] | 36×22.5 | 27, 29 | 6.1 dBi | 1.64, 2.18 | Circular patch with slots | -25 | 2×2 | RT Duroid 5880 | 0.003 | 9.9 |
| [18] | 65×65 | 3–9.3 | 5.1 dB | 1.47 | Decoupling network at ground plane | -15 | 4×4 | FR4 | 0.08 | 9.85 |
| [19] | 65×65 | 2.9–10.86 | 4 dB | NA | Decoupling network and unconnected ground | -22 | 4×4 | LCP | 0.01 | 9.9 |
| [21] | 25×25 | 26.7–36.3 | NA | NA | Hexagonal slotted ground with orthogonal radiator arrangements | -18 | 4×4 | RT 5880 | 0.05 | 9.8 |
| [23] | 75×37.5 | 3.3–5.95 | NA | NA | Parasitic strips and DGS | -15 | 8×8 | FR4 | 0.11 | NA |
| [24] | Radius = 23 mm | 5.9–7.2 | 4.1 dBi | NA | Circular patch with multiple sectored slots | -10 adjacent and -21 dB for others | 8×8 | FR4 | 0.08 | NA |
| This work | 22×17 | 27 (26–28) | 4.64 dB | 0.00349 for .01mW input power | T-shaped patch and slotted array ground for decoupling | -22 dB adjacent, -40 dB for others | 8×8 | FR4 | 0.005 | 9.97 |

## Supporting information

**S1 Dataset. Data set for the graphs is available on, https://docs.google.com/spreadsheets/d/1_nQgtMZ9CdXgqzkAUifkRvG0QytSwH0I/edit?usp=drive_link&ouid=107203764657837180766&rtpof=true&sd=true.**
(XLSX)

## Author Contributions

**Conceptualization:** Anupma Gupta, Shonak Bansal.

**Data curation:** Anupma Gupta, Shonak Bansal.

**Formal analysis:** Mohammed H. Alsharif.

**Funding acquisition:** Peerapong Uthansakul.

**Investigation:** Meet Kumari, Manish Sharma.

**Methodology:** Anupma Gupta, Manish Sharma, Mohammed H. Alsharif, Shonak Bansal.

**Supervision:** Meet Kumari, Manish Sharma, Mohammed H. Alsharif, Peerapong Uthansakul.

**Validation:** Peerapong Uthansakul, Monthippa Uthansakul.

**Visualization:** Monthippa Uthansakul.

**Writing – original draft:** Anupma Gupta, Shonak Bansal.

**Writing – review & editing:** Meet Kumari, Mohammed H. Alsharif, Peerapong Uthansakul, Monthippa Uthansakul.

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
