## [Decision Letter · Decision Letter 0]

4 Jan 2024

PONE-D-23-358542x4 MIMO Antennas at 27 GHz for n261 band and exploring for Body Centric CommunicationPLOS ONE

Dear Dr. Uthansakul,

Thank you for submitting your manuscript to PLOS ONE. After careful consideration, we feel that it has merit but does not fully meet PLOS ONE’s publication criteria as it currently stands. Therefore, we invite you to submit a revised version of the manuscript that addresses the points raised during the review process.

We look forward to receiving your revised manuscript.

Kind regards,

Muhibur Rahman

Academic Editor

PLOS ONE

“This work was supported by Suranaree University of Technology (SUT).”

5. Please provide a complete Data Availability Statement in the submission form, ensuring you include all necessary access information or a reason for why you are unable to make your data freely accessible. If your research concerns only data provided within your submission, please write "All data are in the manuscript and/or supporting information files" as your Data Availability Statement.

6. We note that Figures 17 and 19 in your submission contain copyrighted images. All PLOS content is published under the Creative Commons Attribution License (CC BY 4.0), which means that the manuscript, images, and Supporting Information files will be freely available online, and any third party is permitted to access, download, copy, distribute, and use these materials in any way, even commercially, with proper attribution. For more information, see our copyright guidelines: http://journals.plos.org/plosone/s/licenses-and-copyright.

1. You may seek permission from the original copyright holder of Figures 17 and 19 to publish the content specifically under the CC BY 4.0 license.

Additional Editor Comments:

(1) Please show the novelty in much more comprehensive manner.

(2) According to Reviewer 1, I am also worried: "There is a noticeable discrepancy between the simulation and measurement s-parameters data." Please address it with supporting materials.

(3) More explanation on radiation patterns and radiation performance must be given as recommended by both reviewers.

(4) Finally there must be mathematical analysis with physical support from simulation and measurements.

Reviewers' comments:

Reviewer's Responses to Questions

**Comments to the Author**

1. Is the manuscript technically sound, and do the data support the conclusions?

Reviewer #1: Partly

Reviewer #2: Partly

2. Has the statistical analysis been performed appropriately and rigorously? 

Reviewer #1: Yes

Reviewer #2: N/A

3. Have the authors made all data underlying the findings in their manuscript fully available?

Reviewer #1: Yes

Reviewer #2: Yes

4. Is the manuscript presented in an intelligible fashion and written in standard English?

Reviewer #1: Yes

Reviewer #2: Yes

5. Review Comments to the Author

Reviewer #1: Comments to the Author

1- The gain is significantly lower when compared to the values reported in Table 1. Additionally, it is notably low for millimeter-wave communications at 27 GHz. Please justify.

2- There is a repetition in explaining the findings of the research at the end of the introduction. Please correct.

3- In section II Single Radiator antenna design methodology in free space, the authors mentioned (Antenna is designed using a ring shaped defected ground and a rectangular radiator with two parallel slots). Where is the ring shape in Fig. 1?

4- In Fig. 10(b), the coupling in the opposite patches is evident, potentially exceeding that in Fig. 10(a). Additionally, the coupling between adjacent patches in Fig. 10(b) has increased when compared to MIMO without decoupling technique, as depicted in Fig. 10(a). please justify.

5- There is a noticeable discrepancy between the simulation and measurement s-parameters data. Considering the close proximity of the SMAs in the prototype, have the authors incorporated the SMA model in the simulation?

6- Please incorporate the MIMO performance, ECC, and DG into the comparison in Table 1.

Reviewer #2: The work presented is not novel and many corrections are required to be incorporated in the manuscript.

1. In the title it is mentioned as 2X4 MIMO antenna..in the conclusion section it is mentioned as 2X8 MIMO antenna. It is clear that the authors have no idea of using the MIMO terminology. Authors are asked to clarify what is mXn MIMO antenna system in the manuscript. Also, correct the terminology in the entire manuscript.

2. Table 1 can be given at the end of the last section for better perception to the readers.

3. In the first paragraph of second section, authors have mentioned that a ring shaped DGS is used..but it is not seen in Fig. 1.

4. Representation of S11 and S21 are not correct in the entire manuscript.

5. More explanation on radiation patterns is to be given.

6. PLOS authors have the option to publish the peer review history of their article (what does this mean?). If published, this will include your full peer review and any attached files.

Reviewer #1: No

Reviewer #2: No

---

## [Author Response · Author response to Decision Letter 0]

30 Mar 2024

Dear Editor and Reviewers,

First of all, we would like to take this opportunity to express our sincere thanks to the editor and reviewers for their valuable time to make suggestions to improve the quality of this paper. We have revised the manuscript in accordance to the reviewer’s comments, and the changes in the revised manuscript are highlighted by yellow. In the following, we give a point-by-point reply to the reviewers’ comments:

Editor Comments

Comment1. Please provide a complete Data Availability Statement in the submission form, ensuring you include all necessary access information or a reason for why you are unable to make your data freely accessible. If your research concerns only data provided within your submission, please write "All data are in the manuscript and/or supporting information files" as your Data Availability Statement.

Response1. All data source files are shared as a supplementary file. Result file is also cited in the manuscript as a link with reference number [33].

https://docs.google.com/spreadsheets/d/1_nQgtMZ9CdXgqzkAUifkRvG0QytSwH0I/edit?usp=sharing&ouid=107203764657837180766&rtpof=true&sd=true

Comment2. We note that Figures 17 and 19 in your submission contain copyrighted images. All PLOS content is published under the Creative Commons Attribution License (CC BY 4.0), which means that the manuscript, images, and Supporting Information files will be freely available online, and any third party is permitted to access, download, copy, distribute, and use these materials in any way, even commercially, with proper attribution. For more information, see our copyright guidelines: http://journals.plos.org/plosone/s/licenses-and-copyright.

Response2. Figure 17 and 19 in the manuscript are not taken from any other source. Images are created designed by authors on the software CST Microwave Studio. This software provides a material library for different dielectric materials, bio tissues and 3-D models. 

Comment3. Please show the novelty in much more comprehensive manner.

Response3. New wearable MIMO antennas with solution of all aforementioned issues are of great interest for researchers. Proposed structure has the following features

• Compact and simple geometry

• Stable impedance and radiation characteristics for heterogeneous body tissues, 

• Conformal to use with wearable devices, 

• Ensure tissue safety from EM waves, 

• Excellent diversity performance for MIMO operation. 

• Suitable for both the free space and on-body devices operating at 27 GHz for 5G new radio bands lying in frequency range2 (FR2).

Novelty of antenna is added in the last paragraph of manuscript and highlighted in yellow color.

Comment4. According to Reviewer 1, I am also worried: "There is a noticeable discrepancy between the simulation and measurement s-parameters data." Please address it with supporting materials.

Response4. It is correct that, simulated and measured s-parameters are not overlapped. Measured reflection coefficient has slight wider bandwidth and upward shifted curve. This is due to the number of connectors placed in close proximity and difference in the property of phantom considered in the simulation and measurement. In simulation model all the feedlines are connected with the 50-ohm ports, no SMA model is incorporated in the simulation. Soldering loss, SMA connector effect and phantom property variation has cause to mismatch in s-parameters. Experimental results show that antenna has occupied the required bandwidth for all the radiators from 26 GHz to 28 GHz with good impedance matching and reflection coefficient value below -20 dB. 

• For the good impedance matching reflection coefficient value should be below -10 dB, here all the measured values are below -30 dB (Figure 21a). Maximum variation of simulated and measured reflection coefficient is 5 dB.

• Impedance bandwidth for simulated antenna is 2 GHz (26 to 28 GHz), for the measured results it is about 2.2 GHz (25.9 GHz to 26.1 GHz).

• For MIMO performance port isolation should be lower than -15 dB, here for all the measured and simulated plots it is below -22 dB (Figure 21 b). Over the operating bandwidth, variation of 2 dB is observed for the simulated and measured results.

• Similar type of variation in simulated and experimental results is found in various literature also.

[1]. Gao GP, Hu B, Wang SF and Yang C (2018) Wearable circular ring slot antenna with EBG structure for wireless body area network. IEEE Antennas and Wireless Propagation Letters 17, 434–437.

[2]. Hu B, Gao GP, He LL, Cong XD and Zhao JN (2017) Bending and on-arm effects on a wearable antenna for 2.45 GHz body area network. IEEE Antennas and Wireless Propagation Letters 15, 378–381

[3]. Koo T, Hong Y, Park G, Shin K and Yook J (2015) Extremely low-profile antenna for attachable bio-sensors. IEEE Transactions on Antennas and Propagation 63, 1537–1544.

Comment5. More explanation on radiation patterns and radiation performance must be given as recommended by both reviewers.

Resposne5. As suggested radiation pattern is explained more and directivity plot is also added to show the radiation intensity orientation.

Figure 22 shows the 2-D radiation pattern of antenna for free space and on body operating conditions at 27 GHz. Dipole-like radiation pattern has been observed for free space and broadside radiation pattern is observed for on-body condition in both the E-plane and H-plane. On-body operation, internally reflected and refracted waves superimposed with the antenna radiations and increases the forward radiation of the antenna. Broadside radiation pattern is desirable for the on body devices for making reliable communication link with the external monitoring devices and reducing the absorption of EM waves by body tissue. It is virtually impossible to maintain same orientation with on and off body device due to different human activities. Therefore, broadside radiation pattern helps to overcome orientation mismatch and transmits maximum energy in the direction opposite to body. Simulated and measured results are in close approximation to each other. Both in E-plane and H-plane maximum radiation intensity can be observed at 300 and 3150. 3-D power pattern for the antenna is shown in Figure 23. Maximum free space gain is 2.53 dB and on-body gain is 4.64 dB. As back radiations are reduced in the on-body operations (Figure 22), antenna gain is improved in forward direction. Reflection coefficient of proposed structure is also improved for on-body configuration and justify the antenna on-body performance. Besides this, directivity of antenna is one of the important characteristic for the on-body devices. Figure 24 shows the 3-D directivity plot for the free space and on-body operation and confirming a forward-oriented beam suitable for on body to off body communications. Maximum directivity of 5.06 dBi and 9.5 dBi is achieved for free space and on-body conditions respectively. High directivity shows that antenna radiation intensity is higher in a specific direction. Maximum radiations are oriented towards the normal to the surface of body tissue, thus reducing the absorption and loss of radiations in body tissue. Improvement of directivity for on-body condition is due to internal reflections by the heterogeneous body tissue layers. 

(a) (b)

Figure 24 3-D directivity plot (a) free space for port-2 (b) free space for port-8 (c) on-body port-2 (d) on-body port 8 

Reviewers' comments:

Reviewer #1: Comments to the Author

Comment1. The gain is significantly lower when compared to the values reported in Table 1. Additionally, it is notably low for millimeter-wave communications at 27 GHz. Please justify.

Resposne1. In is studied that directivity is the important function for the body area network application. Broadside unidirectional radiation id desired for on body to off body communication. Comparison table is corrected and directivity is compared with the directivity of the existing structures at 27 GHz. To justify the gain value Figure 24, representing the 3-D directivity plot of antenna at 27 GHz is shown. Proposed structure has more than 9.5 dBi gain in broadside direction which is better than the other MIMO structures reported in comparison Table.

Figure 24 3-D directivity plot (a) free space for port-2 (b) free space for port-8 (c) on-body port-2 (d) on-body port 8 

Comment2. There is a repetition in explaining the findings of the research at the end of the introduction. Please correct.

Response2. The repeated text at the end of Introduction section is removed. 

Comment3. In section II Single Radiator antenna design methodology in free space, the authors mentioned (Antenna is designed using a ring shaped defected ground and a rectangular radiator with two parallel slots). Where is the ring shape in Fig. 1?

Response3. As the shape of the ground plane is a rectangular closed shaped, it is corrected as rectangular-shaped ring ground.

Comment4. In Fig. 10(b), the coupling in the opposite patches is evident, potentially exceeding that in Fig. 10(a). Additionally, the coupling between adjacent patches in Fig. 10(b) has increased when compared to MIMO without decoupling technique, as depicted in Fig. 10(a). please justify.

Response4. Without the decoupling network, in 4-port MIMO all the radiators (opposite and adjacent) lost the resonance frequency and detunes at 30 GHz as shown in Figure 10 (S-parameter plot for 4-port MIMO without decoupling). Thus, the less coupling is visible in the surface current plot in Figure 11 (a) at 27 GHz. After adding the decoupling network, all the radiators are tuned at 27 GHz, and coupling effect for the same frequency is visible in Figure 11 (b). To justify this three surface current distribution plots are shown. In 2-port MIMO and in 4-port MIMO antenna with decoupling network, adjacent radiators have similar coupling effect. 

 Figure 7 (c) 2-port MIMO without decoupling Figure 11 (a) Figure 11(b)

Figure 11. Surface current distribution of 4-element structure (a) without decoupling structure (b) with decoupling structure

S-parameters plot for all the three configurations are also compared and confirms the isolation of antenna. In Figure 8, S-parameter plot for 2-port antenna. Here the isolation between two adjacent radiators is -22.5 dB.

Figure (10), S-parameter plot for 4-port MIMO antenna without decoupling network shows that frequency of patch1(S11) and patch2(S22) is detuned at 30 GHz (adjacent patches), and opposite patches are not resonating (S33) and (S44). Thus, the S21, S31, and S41 are showing high isolation. Due to this frequency detuning effect surface current plot without decoupling network shows less coupling at 27 GHz in adjacent radiator. In order to make antenna resonance frequency and impedance matching stable and avoid the coupling effect a decoupling network is added. 

S-parameter plots for 4-port antenna after decoupling network (Figure 14) shows that isolation between patch1 and patch2 is -21 dB which is slight less than the port-2 configuration.

Commnet5. There is a noticeable discrepancy between the simulation and measurement s-parameters data. Considering the close proximity of the SMAs in the prototype, have the authors incorporated the SMA model in the simulation?

Response5. It is correct that, simulated and measured s-parameters are not overlapped. Measured reflection coefficient has slight wider bandwidth and upward shifted curve. This is due to the number of connectors placed in close proximity and difference in the property of phantom considered in the simulation and measurement. 

In simulation model all the feedlines are connected with the 50-ohm ports, no SMA model is incorporated in the simulation. Soldering loss, SMA connector effect and phantom property variation has cause to mismatch in s-parameters. Experimental results show that antenna has occupied the required bandwidth for all the radiators from 26 GHz to 28 GHz with good impedance matching and reflection coefficient value below -20 dB. Similar type of variation in simulated and experimental results is found in various literature also.

[1]. Gao GP, Hu B, Wang SF and Yang C (2018) Wearable circular ring slot antenna with EBG structure for wireless body area network. IEEE Antennas and Wireless Propagation Letters 17, 434–437.

[2]. Hu B, Gao GP, He LL, Cong XD and Zhao JN (2017) Bending and on-arm effects on a wearable antenna for 2.45 GHz body area network. IEEE Antennas and Wireless Propagation Letters 15, 378–381

[3]. Koo T, Hong Y, Park G, Shin K and Yook J (2015) Extremely low-profile antenna for attachable bio-sensors. IEEE Transactions on Antennas and Propagation 63, 1537–1544.

Commnet6. Please incorporate the MIMO performance, ECC, and DG into the comparison in Table 1.

Response6: As suggested by the reviewer Columns for ECC and DG parameters are included in the comparison table in the manuscript.

Reviewer #2: Comments to the Author

Comment1: In the title it is mentioned as 2X4 MIMO antenna. in the conclusion section it is mentioned as 2X8 MIMO antenna. It is clear that the authors have no idea of using the MIMO terminology. Authors are asked to clarify what is mXn MIMO antenna system in the manuscript. Also, correct the terminology in the entire manuscript.

Response1: Authors are thankful for this valuable comment. MIMO mXn is corrected in the entire manuscript and highlighted in yellow color.

Comment2: Table 1 can be given at the end of the last section for better perception to the readers.

Response2: Table 1 is now included in the end of the last section of the manuscript. 

Comment3: In the first paragraph of second section, authors have mentioned that a ring shaped DGS is used. but it is not seen in Fig. 1.

Response3: As the shape of the ground plane is a rectangular closed shaped, It is corrected as rectangular-shaped ring ground.

Comment4: Representation of S11 and S21 are not correct in the entire manuscript.

Response4: As suggested by reviewer S11 AND S21 is written correctly in the manuscript.

Commnet5: More explanation on radiation patterns is to be given.

Response5: As suggested radiation pattern is explained more and directivity plot is also added to show the radiation intensity orientation.

Figure 22 shows the 2-D radiation pattern of antenna for free space and on body operating conditions at 27 GHz. Dipole-like radiation pattern has been observed for free space and broadside radiation pattern is observed for on-body condition in both the E-plane and H-plane. On-body operation, internally reflected and refracted waves superimposed with the antenna radiations and increases the forward radiation of the antenna. Broadside radiation pattern is desirable for the on body devices for making reliable communication link with the external monitoring devices and reducing the absorption of EM waves by body tissue. It is virtually impossible to maintain same orientation with on and off body device due to different human activities. Therefore, broadside radiation pattern helps to overcome orientation mismatch and transmits maximum energy in the direction opposite to body. Simulated and measured results are in close approximation to each other. Both in E-plane and H-plane maximum radiation intensity can be observed at 300 and 3150. 3-D power pattern for the antenna is shown in Figure 23. Maximum free space gain is 2.53 dB and on-body gain is 4.64 dB. As back radiations are reduced in the on-body operations (Figure 22), antenna gain is improved in forward direction. Reflection coefficient of proposed structure is also improved for on-body configuration and justify the antenna on-body performance. Besides this, directivity of antenna is one of the important characteristic for the on-body devices. Figure 24 shows the 3-D directivity plot for the free space and on-body operation and confirming a forward-oriented beam suitable for on body to off body communications. Maximum directivity of 5.06 dBi and 9.5 dBi is achieved for free space and on-body conditions respectively. High directivity shows that antenna radiation intensity is hi

---

## [Editor Report · Decision Letter 1]

13 May 2024

PONE-D-23-35854R18-port MIMO Antenna at 27 GHz for n261 band and exploring for Body Centric CommunicationPLOS ONE

Dear Dr. Uthansakul,

Thank you for submitting your manuscript to PLOS ONE. After careful consideration, we feel that it has merit but does not fully meet PLOS ONE’s publication criteria as it currently stands. Therefore, we invite you to submit a revised version of the manuscript that addresses the points raised during the review process.

We look forward to receiving your revised manuscript.

Kind regards,

Muhibur Rahman

Academic Editor

PLOS ONE

Journal Requirements:

Additional Editor Comments:

Commnet5. There is a noticeable discrepancy between the simulation and measurement sparameters data. Considering the close proximity of the SMAs in the prototype, have the authors

incorporated the SMA model in the simulation?

New comment: Incorporate SMA connector and compare with measured data?

There are so many unnecessary references that need to be eliminated. Editor is highlighting that please eliminate even if the reviewers recommended. It is completely irrelevant.

Like:

[2]. Alsharif M. H., Jahid A., Kannadasan R., Kim M.-K., Unleashing the potential of sixth generation (6G)

wireless networks in smart energy grid management: a comprehensive review, Energy Rep. 11 (2024)

1376–1398.

[3]. Alsharif M.H., Alsharif Y.H., Chaudhry S., Albreem M.A., Jahid A., Hwang E., Artificial intelligence

technology for diagnosing COVID-19 cases: a review of substantial issues, Eur. Rev. Med. Pharmacol. Sci.

24 (2020) 9226–9233.

[4]. Alsharif M. H., Hossain M., Jahid A., Khan M., Choi B.J., M Mostafa S., Milestones of wireless

communication networks and technology prospect of next generation (6G), Comput. Mater. Continua

(CMC) 71 (2022) 4804–4818.

[5]. menroodaki, W. Haines, M. Fromandi, and Z. Popovic, ‘‘Non-invasive internal body temperature tracking

with near-field microwave radiometry,’’ IEEE Trans. Microw. Theory Techn., vol. 66, no. 5, pp. 2535–

2545, May 2018.

[15]. A. Iqbal, A. Basir, A. Smida et al., “Electromagnetic bandgap backed millimeter-wave MIMO antenna for

wearable applications,” IEEE Access, vol. 7, pp. 111135–111144, 2019.

[6]. Alsharif M.H., Comparative analysis of solar-powered base stations for green mobile networks, Energies 10

(2017) 1208.

---

## [Author Response · Author response to Decision Letter 1]

29 May 2024

The response to reviewer comments is given in the attached file.

---

## [Editor Report · Decision Letter 2]

2 Jun 2024

8-port MIMO Antenna at 27 GHz for n261 band and exploring for Body Centric Communication

PONE-D-23-35854R2

Dear Dr. Peerapong Uthansakul,

We’re pleased to inform you that your manuscript has been judged scientifically suitable for publication and will be formally accepted for publication once it meets all outstanding technical requirements.

Kind regards,

Muhibur Rahman

Academic Editor

PLOS ONE

Additional Editor Comments (optional):

I have no further concerns. The manuscript can be published in current form however careful modification is needed to make in the format of PLOS one.
---

## [Editor Report · Acceptance letter]

7 Jun 2024

PONE-D-23-35854R2 

PLOS ONE

Dear Dr. Uthansakul, 

I'm pleased to inform you that your manuscript has been deemed suitable for publication in PLOS ONE. Congratulations! Your manuscript is now being handed over to our production team.

Kind regards, 

on behalf of

Professor Muhibur Rahman 

Academic Editor

PLOS ONE